# Implementation of Smart Infusion Pumps: A Scoping Review and Case Study Discussion of the Evidence of the Role of the Pharmacist

**DOI:** 10.3390/pharmacy8040239

**Published:** 2020-12-16

**Authors:** Neha Shah, Yogini Jani

**Affiliations:** 1Centre for Medicines Optimisation Research and Education, University College London Hospitals Foundation Trust, London NW1 2BU, UK; 2The School of Pharmacy, University College London, London WC1N 1AX, UK

**Keywords:** pharmacist, drug library, drug error reduction software, smart infusion pump, smart pump, implementation

## Abstract

“Smart” infusion pumps include built in drug error reduction software which uses a drug library. Studies have reported the drug library build should be undertaken by a multidisciplinary team, including a pharmacist; however, the extent or nature of the input required by the pharmacist for greatest benefit is unknown. This review aimed to identify key factors for the implementation of the smart infusion pumps, with a focus on the role of pharmacists and compare this to the experience from a case study. A literature review was conducted using Embase and Ovid Medline, and 13 eligible papers were found. Predominant themes relating to the pharmacist’s role and successful implementation of the smart infusion pumps were determined. Key factors for success included team involvement across the entire process from procurement, set-up through to implementation including risk assessment and device distribution, and training, which were comparable to the case study experience. Few studies described the extent or details of the pharmacist’s responsibilities.

## 1. Introduction

The National Patient Safety Agency (NPSA) in England identified that incidents involving injectable medicines represented 62% of all reported incidents leading to death or severe harm [1]. Observational studies conducted at multiple hospitals within the United Kingdom (UK) and the United States (US) have reported error rates of up to approximately 60% [2,3], with recent studies reporting that only up to 1.1% of all the discrepancies and infusion errors witnessed were potentially harmful, and none would have prolonged hospital stay or resulted in long term harm [2,4].

Infusion pumps that control the administration of medication infusions at a set rate were introduced worldwide over 40 years ago. These have evolved vastly over time from manual control into sophisticated automated systems with in-built safety features and ability to interface with other electronic systems [5]. Safety features include a sensor for detecting air within the line; a battery fail safe system, which is a backup power system and ensures that the infusion pumps continue administering medication in the event of an electricity outage; pressure sensors which can detect pressure changes within the lines cause by issues such as blocked lines or veins or empty bags of medication. The newer, so called “smart” infusion pumps include built in drug error reduction software (DERS), which uses a bespoke drug library. DERS is considered a safety feature as it has the capacity to have pre-set limits for specific medications, thereby preventing unsafe infusion rates or errors in dose calculation [5].

Benefits of smart infusion pumps and the DERS are dependent on software, the drug library set-up and limits as well as uptake and use in practice [6]. The uptake of the smart pumps with incorporated DERS has been higher in the US and Canada compared to the UK [2,5]. A study of 120 UK hospitals found that majority of the hospitals do not use DERS, and of the ones that did, use was limited for certain drugs or for certain high-risk clinical areas, with critical care being the most common [2,4,7]. Due to the infusion pump manufacturers only producing smart infusion pumps now, the uptake will increase throughout the UK. Coupled with the World Health Organisation’s (WHO) third patient safety global challenge to reduce harm caused by medication errors, this may result in an increase in the use of the DERS software across the UK [8]. Reported benefits of using the DERS to prevent harm include the interception of errors such as wrong rate, wrong dose, pump setting errors, reduced adverse drug reaction rates, improvements in practice and cost effectiveness [5,9,10,11,12].

However, it is recognised that as with the introduction of any new technology, not all errors will be eliminated, and new errors may occur [4,5]. A multisite study conducted in the UK revealed that using smart pumps could decrease errors caused by gravity infusions (i.e., not using any technology to limit the rate of infusion); however, new errors or unintended consequences could be introduced when using newer technology such as mis-selection errors of the drug library entry [13]. Another study of 29 hospitals in Canada showed that the potential safety benefits of the smart pumps were not being realised due to multiple failures during the implementation stages [6]. These studies emphasise the importance of the set-up of DERS and successful implementation in realising the error reduction benefits. In England, pharmacists are recognised as medicines experts and have an active role in improving medication safety. Examples of pharmacist led initiatives to reduce the risk of injectable medicines include standardisation of medication infusions for the intensive care unit [14] and a national injectable medicines guide [15].

Our hypothesis was that whilst studies highlight that the drug library build should be undertaken by a multidisciplinary team, including a pharmacist, few detail the extent or nature of the input required by the pharmacist for greatest benefit. Therefore, the primary aim of this paper was to review the literature to identify key factors for the implementation of the smart infusion pumps, with a focus on the role of pharmacists; the secondary aim was to discuss a case study of building and implementing a drug library at an acute care tertiary academic hospital in London, in the context of the review findings.

## 2. Method

A scoping literature review was conducted using Embase and Ovid Medline (all from inception to 13 July 2020). A review protocol was developed by the authors but not registered. The same search strategy was used for all the databases, using key words as follows: (“smart pump” OR “smart pumps” OR “intravenous smart pumps” OR “smart IV pump” OR “smart pump safety” OR “smart pump technology”) AND (“drug libraries” OR “drug library” OR “drug error reduction software”) AND (“pharmacist” OR “pharmacy”) AND (“implement*” and “develop*”). Articles were restricted to the English language. All duplicates were removed.

Inclusion criteria were any original research studies, narrative papers or abstracts detailing or describing the process of implementation of the smart pumps or the process of creating the drug library. All other studies including those reporting the impact of the smart infusion pumps without details of implementation strategy or drug library set-up were excluded.

Titles and abstracts were screened against the eligibility criteria, and for those which may be eligible, the full texts were reviewed. References of included articles were also screened for other potentially relevant titles. All screening activities were carried out by the lead author, and descriptive information including the study objective, reported intervention, reported study outcomes, study setting, whether standardised concentration infusions were used and details of the healthcare professionals involved in the implementation and development of the drug library was collated into a table.

Themes that featured prominently in the included literature were identified using a deductive thematic approach and compared to the experience in a case study from the perspective of the pharmacy team at a tertiary care academic hospital. The implementation team at the case study site consisted of medical device experts, IT experts, nursing staff, consultants, finance department, education leads within nursing, the medication safety officer and pharmacists. The latter were tasked with the responsibility of creating a drug library that would be suitable for use across the entire organisation.

## 3. Results

A total of 44 papers were identified using the search criteria, which reduced to 26 when duplicates were removed and limited to “English language”. Of these, thirteen papers met the eligibility criteria. The rest were not eligible as these did not specifically describe steps taken to build the drug library or describe the process of implementation of smart pumps. From screening the references of the included and excluded studies, no further studies were found.

All the included articles involved pharmacists in the development of the drug library (Table 1); nine involved nursing staff and ten had at least one physician involved with the development of the drug library. Four of the studies had implemented smart infusion pumps throughout the organisations [16,17,18,19]. Six of the studies had implemented smart infusion pumps only within paediatric intensive care or in a paediatric setting only [10,20,21,22,23,24]. Five of the studies had developed drug libraries for other specific areas, including intensive care units, ambulatory settings, for home infusions, for epidurals only or for anti-infectives only [12,25,26,27,28].

Key factors for success, as identified from themes of the literature, included team involvement across the entire process from procurement, set-up through to implementation including risk assessment and device distribution, and training. Each key theme is discussed further with reference to our case study.

### 3.1. Procurement

Two studies discussed the importance of ensuring that all the correct parties are involved in the procurement of the smart infusion pumps [20,27]. One study included representation from pharmacy, as well as clinicians, nursing, biomedical engineering, purchasing and human factor experts [27] and the other included nursing, pharmacy, clinical engineering and physicians [20]. This enabled all parties to contribute to the selection of the pump to be purchased, and to have an awareness of any strengths and limitations of the smart pumps, which could be factored into the drug library build [20,27]. This is similar to the case study, where procurement was led by the medical physics team and involved nurses and a medication safety pharmacist.

Our experience shows similarities to the literature. The implementation of the smart infusion pumps was led by the medical physics team during an organisation-wide project to replace the older infusion pumps, and all relevant parties, including nursing staff, physicians, the medication safety officer and pharmacists were invited to the procurement presentations during the tendering process. As with all technology, smart infusion pumps were noted to have some limitations; however, the limitations vary with different manufacturers. Knowledge of the limitations was of utmost importance, as this had a direct effect on how the drug library was built. Procurement of the smart infusion pumps was based on a scoring system and considered the potential impact of change from the infusion pumps already used (as recommended by end-users, mainly nursing staff), aesthetics of the pumps including size and display of infusion details to take into account human factors, as well as cost associated with the purchase of new pumps.

### 3.2. Implementation Strategy

#### 3.2.1. Risk Assessment

Two studies used a human factors approach prior to implementing the smart infusion pumps using models such as “failure mode and effect analysis” (FMEA) [21,28]. The identification of risk points in the different stages of implementation should be carried out by a multidisciplinary team, including pharmacists as demonstrated by Manrique et al. [21] and Wetterneck et al. [18]. This is an important step as appropriate action can be taken to manage the identified risks during the implementation stage. In our case, whilst a formal human factors assessment was not conducted at the case study site, a hazard workshop was conducted, and a clinical safety case developed [19].

#### 3.2.2. Device Distribution

Unlike the case study where the intention was to introduce smart infusion devices across all clinical areas for all medicines, the majority of the studies described smart infusion pump implementation in specific units, such as intensive care settings where high-risk infusions are often administered or for a particular set of drugs, such as epidurals or anti-infectives [10,20,21,22,23,24].

This required decisions about the drug library build and visibility. With approval and input from relevant clinical governance committees within the organisation, a single drug library was created, with each of the medication infusions configured to appear in either all care setting areas or in specific ones only. This approach was considered the most suitable one, due to the flow of the patients within the organisation who may have infusions in situ whilst being transferred from one care setting to the next—for example, from a theatre recovery setting to the critical care unit. Another benefit of this was that all care area entries could be accessed by any care area in case of any outlying or transitional patients—for example, requiring initiation of critical care infusions in a ward environment whilst awaiting transfer to a critical care unit or vice versa.

#### 3.2.3. Drug Library Creation

The two predominant themes for the creation of a drug library were using a multidisciplinary team, consisting of a pharmacist, nurses and physicians [10,12,17,20,22,23,24,25,26], and standardisation of infusions [10,17,18,19,20,21,22,23,26,27,28].

The approach and time taken varied considerably. For example, Howett et al. described the creation of a drug library was often the rate limiting step in the implementation of smart infusion pumps [22]. Manrique-Rodriguez et al. described creating the drug library for a paediatric intensive care unit with a multidisciplinary team consisting of a clinical pharmacist, a physician and the chief nurse of the unit over a period of seven months. Literature reviews were undertaken to determine standard concentrations and accurate limits for intravenous infusions for high-risk drugs [10,11]. Another group standardised the process for managing smart pump drug libraries within a specific geographical area by forming a working group comprising pharmacists, nurses and industrial engineers, with the main tasks being to evaluate the variability of different drug libraries being used and advice on standardised dosing limits, alerts, policies and best safety practices [17].

The case study site used a mixture of the two approaches. In order to ensure that the drug library was of a robust build, the strengths and limitations of the specific DERS were considered using a risk based approach and focussing on high-risk medicines. A multidisciplinary clinical advisory group, consisting of medical device experts, consultants, pharmacists and nursing staff from across the organisation provided input on the drug library development, taking into consideration any professional practice, governance and safety implications. Principles and processes were established to ensure a standardised approach for the drug library build and maintenance. These principles were applied to mitigate some of the foreseen errors.

The creation of the drug library required practices from across the organisation to be standardised to a certain extent—i.e., the same infusion should be administered in the same way in order to have the same limits for the drug for the limits to have meaning. In the creation of the drug library, all the different disciplines were approached and questions were raised when issues in practice were uncovered or when standardisation could not be achieved. For the latter specific drug library entries were made for the different uses, and careful consideration was applied when considering which care setting was selected to ensure mis-selection errors were minimised. Clinical pharmacists from differing specialties were asked to work within their multidisciplinary teams, consisting of nursing staff and clinicians from specialist areas to help standardise infusion practice across the organisation.

The pre-existing infusion pumps required the rate to be set manually in millilitres/hour (mL/h). The drug library set-up allowed the display to default to rate (mL/h) or dose (mg/kg/min). The clinical advisory group was consulted about this as it required a change in work practices and introduced a potential risk in misinterpreting the display value. The group advised that, for safety reasons, the dosing units and rate display should match the prescription as much as possible.

In our experience, the drug library creation took place over a period of approximately 18 months and was led by the medication safety team within the pharmacy, with input from a multidisciplinary clinical advisory group and oversight from the hospital medication safety committee. This was a longer time frame than was originally anticipated due to the standardisation process and checking/governance processes implemented each time the drug library was amended. None of the literature stated exactly what governance processes had been implemented for any of the sites that used drug libraries.

#### 3.2.4. Training

The literature from the search did not mention whether training the end-users on how to use smart infusion pumps should involve a pharmacist or not. A quality improvement study showed that continuous education on the use of smart infusion pumps can increase the compliance of its use [29].

In the case study, a nurse was appointed to lead on education and training of the smart infusion pumps. In our opinion, it was imperative that the end-users were all trained appropriately as all workflows relating to setting up an infusion were to change due to the use of the drug library. Education of the end-users should be based on their speciality, as certain drugs may need more information being input than others, and re-education should be encouraged if needed. End-users should also have dedicated time in a “play” environment where they can experiment with the pumps using the drug library and ask questions that may arise. This may help reduce workarounds being used later on and will help to ensure that the pumps are being used in a standardised way. From our experience, a pharmacist should be involved in the training or be involved in setting up training tools as well as they have valuable knowledge specifically about the drug library and how that will work. Ensuring the correct training tools are used for each specific clinical setting can minimise errors after the pumps are implemented and can help keep patients safe.

## 4. Discussion

There are few publications that describe the key factors for implementation of smart infusion pumps or detail the composition of teams and the process for drug library set-up. However, the limited literature and our experience shows that involvement of a multidisciplinary group, including pharmacists, is essential from the purchasing process through to implementation to facilitate a clear strategy for drug library build and subsequent adoption and use.

The majority of the literature states that the drug library was created with a multidisciplinary team consisting of consultants, nursing staff and pharmacists. However, a pharmacist within in team was constant in all the included studies. Our experience was that the drug library build was led by the pharmacy that had overall responsibility for it and had input from different healthcare professionals. Pharmacists, as medicines experts, are best placed to create the drug library, as they have the working knowledge of how these medicines are used, the evidence behind their use, and the concentrations required. Advising on how medications should be prepared and administered is inherent to a pharmacist’s role and therefore they have the necessary expertise and competencies. Having input from end-users (nursing staff) gives important input into how the drug library may change the workflow for how to administer medications for certain medications. For some medications, the workflow may become more complicated, and for other medications the workflow may become easier. Consultant input from all specialities is also important, as the way the drug library may be set-up for a particular drug may require a change in how the drug is prescribed—e.g., instead of the rate (mL/h) being present within the prescription, the dose (mg/kg/min) may need to be prescribed. Furthermore, consultants were in a position to say when the drug may be used over the normal limits, and this helped to design dosing limits.

There is evidence that infusions should be standardised as much as possible. Our experience is that this is very difficult to do with certain drugs, as some units use a higher concentration due to the required response. Some of the drugs were standardised within our organisation; however, there was a wide variety of drugs that could be used in a number of concentrations depending on the clinical situation of the patient and the rate would greatly vary depending on the final volume to be infused. The lack of standardisation for the use of an opiate drug was one of the reasons that an incident occurred whereby a patient received an overdose of opioids. In this example, if the concentration had been standardised for use across the organisation, the concentration which was incorrectly input manually would not have been required, and the correct bolus dose would have been administered. Furthermore, standardised infusions across a geographical area or a nation would provide a safety feature in terms of the end-users being familiar with the concentrations and how to administer them. If standardisation did occur, the pharmaceutical industry may be inclined to form premade infusions at those concentrations meaning that the nursing staff would not need to manipulate the drugs during the reconstitution phase of infusion preparation to obtain required concentrations.

One paper specified how long the drug library took to create (7 months for a stand-alone paediatric intensive care unit) [10,11]. From our experience, it is essential to have realistic time expectations for a dedicated pharmacist to create a drug library, with collaboration from the multidisciplinary team. Creating a comprehensive robust drug library which may be useful in preventing errors will take time, and from our experience took more than a year. In the literature, one of the reasons why the compliance rates to using the drug library was low was due to the drug library not being updated frequently [29]. The time required for maintenance of the drug library should be thought about when putting in a financial bid to procure smart infusion pumps to ensure that the drug library is maintained, and therefore remains useful. Unfortunately, we were not able to measure the compliance of the use of drug libraries post implementation.

As with all introduction of new technology, some unintended consequences or errors may occur [13]. Studies showed that using smart infusion pumps alone will not prevent all infusion errors; however, using a closed loop system where the smart pumps and the electronic prescribing systems communicate with one another may prevent a larger proportion of errors [4,16,30]. This was demonstrated in a study by Gerhart et al. whereby the smart infusion pumps were used as an integrated system with the electronic health records, and demonstrated a high compliance rate of 97% and 782 significant errors were prevented [16]. Ohashi et al. also described some of the negative effects of implementing smart infusion pumps. Some of these were lower compliance rates of using smart pumps, soft alerts being over-ridden, not all errors being intercepted, or the possibility of using the wrong drug library [5]. A continuous quality improvement study showed that compliance to using the drug library varied between different clinical areas, and constant education, audits and drug library refinement after implementation led to increased compliance [29]. This all shows that the workflows for using the smart infusion pumps are vastly different from traditional pumps as these did not incorporate drug libraries.

We used a scoping review to gain an overview of the evidence of the role of the pharmacist in implementing smart infusion pumps. The literature search showed pharmacists were key stakeholders in the development of the drug library and in the implementation of the smart infusion pumps, as having a pharmacist during the development and implementation was a constant [10,12,16,17,18,20,21,22,23,24,25,26,27,28,31]. The presumed reason for this being that pharmacists are recognised as medication experts and would be in the best position to create a drug library. Our experience has shown that pharmacists have a pivotal role in ensuring an accurate and safe drug library is built, which would be of utmost importance when it comes to patient safety. A limitation of the scoping review approach is that it cannot not provide answers to specific questions. For example, in this review the evidence of the exact role and impact of pharmacists’ involvement in specific areas such as training cannot be synthesised.

The main learning points from the literature and our experience was that the drug library build should be of robust quality, and appropriate governance processes should be followed. A multidisciplinary team should be involved in all stages of the drug library build from procurement to actually testing and implementing the pumps. Infusion pumps have been used for years; however, drug libraries have not, and this changes the workflow for the administration of medication. From our perspective, as the pharmacist had the main role of creating a drug library, the training should also include the pharmacist/the team responsible for the build as they may be able to tailor the training to specific care areas and drugs frequently used in these care areas. Other points to take into account are to ensure that enough time is given to build a comprehensive drug library, and that drugs which are used across many specialities are standardised. The standardisation process of the drugs may need to go through organisation-wide committees, such as the Drug and Therapeutics Committee. Some drugs may not be able to be standardised due to the clinical need, and these drugs should have appropriate risk assessments conducted to minimise errors from occurring.

## 5. Conclusions

With the implementation of any new health technology, the organisations should involve the key stakeholders. In the case of smart infusion pumps, pharmacists have a role at all stages—from the tendering and procurement process through to implementation and evaluation. Standardisation of intravenous infusions should be considered a prerequisite to building a drug library for smart infusion pumps, and sufficient time and resources should be allocated for the creation of the drug library in order for it to be accurate and safe.

## Figures and Tables

**Table 1 pharmacy-08-00239-t001:** Summary of articles reviewed.

Author and Year of Publication	Objective	Intervention	Outcomes	Setting	Standardised Concentration Infusions Used?	Who Was Involved in the Development of the Drug Library?
Brown T.D. et al., 2018 [26]	Smart pump technology not available for home infusions	Creation of a drug library for home infusion providers	Successfully implemented a drug library for home infusions.	Hospital based and rural based home infusion providers	Not stated	Three clinical pharmacists at one site; two nurses and one pharmacist at another site
Butler E. et al., 2013 [23]	Use standardised concentration infusions	Compile a drug library of standard concentrations drugs to be administered on paediatric intensive care unit	Standard concentration switch was successfully implemented.	Paediatric intensive care	Yes	Paediatric intensive care unit consultant and pharmacist.
Chuk A.C. et al., 2012 [19]	Quality improvement (QI) project to administration of intravenous medicines	QI review of the pumps DERS data enabled further optimisation of the drug library	DERS limits within the drug library were able to be added or optimised based on medication use throughout the centre	Academic medical centre	Yes	Pharmacy and therapeutics drug library subcommittee (involves pharmacists, nurses, analysts and patient safety staff).
Delage E. et al., 2012 [28]	Evaluate feasibility of including anti-infectives with useful limits and evaluate user satisfaction	Develop a drug library for anti-infective medicines	It is feasible to integrate anti-infectives with useful limits into the drug library if there is a standard concentration and administration times are standardised.	Anti-infectives only at a mother and child university hospital centre	Yes	Pharmacists and physicians
Dimech A. et al., 2012 [12]	Implement smart pumps in the intensive care unit (ICU) to aid safer drug administration	Develop a drug library and implement smart pumps	Drug errors reduced; the design of the drug library was sensitive enough to ensure safe drug administration and was practical enough to enable consistent use of the system.	Intensive care unit	Not stated	ICU pharmacist and consultant intensivist.
Gerhart et al., 2013 [16]	Describing implementation of intravenous clinical integration (IVCI)	Implementing intravenous clinical integration	A 27% reduction in nursing time achieved with the use of the integrated system when starting and documenting each new infusion. Numerous steps in the process of manually programming the pumps are eliminated with the IVCI process.	10 outpatient health centres and 3 hospitals	No	Biomedical engineering, pharmacy and nursing
Howlett M. et al., 2016 [22]	Standardising drug concentrations	Expansion of a drug library that was set-up with standardised concentration infusions in a paediatric hospital	Successful amendments and extension to the original drug library	Paediatrics hospital	Yes	Multidisciplinary with pharmacy input
Kennerly J. et al., 2012 [25]	Describes the experience with implementation of smart infusion pumps for epidurals	Develop a drug library for all epidural infusions and update the order sets within the computerised prescriber order entry (CPOE) system	Limited data postimplementation as used keystroke recordings but need for education highlighted. Smart pumps do not ensure improved patient care.	Epidural infusions only at an 803-bed academic medical centre	Not stated	Clinical pharmacist, physicians and nurses
Larsen G.Y. et al., 2005 [20]	Combining standard strength concentrations with smart pump technology reduced reported medication infusion errors	Adoption of standard drug concentrations, implementation of smart syringe pumps and medication label re-engineering	A 73% reduction in the number of reported errors associated with continuous medication infusions. Preparation errors that occurred in the pharmacy decreased from 0.66 to 0.16 per 1000 doses. The number of 10-fold errors in dosage decreased from 0.41 to 0.08 per 1000 doses.	242-bed tertiary paediatric hospital	Yes	Nursing, pharmacy, clinical engineering, physicians (neonatologist, paediatric intensivist, cardiothoracic surgeon, and anaesthesiologist) and the safety manager for the hospital.
Manrique-Rodriguez S. et al., 2014 [21]	Identify risk points in different stages of the smart infusion pumps implementation process	Failure modes and effects analysis carried out preimplementation and post implementation of smart pumps—to identify actions for improvement.	Appropriate risk assessments made it possible to ensure risks are managed during implementation.	Paediatric intensive care	Not stated	Two intensive care paediatricians, two clinical pharmacists and a nurse manager, with a pharmacist being responsible for the whole process
Manrique-Rodriguez S. et al., 2014 [24]	Cost effectiveness of smart pump technology	Development and implementation of a drug library and analysis of reports of intercepted errors	Smart pump technology implementation is cost effective. An estimated EUR 172,279 were saved by prevented adverse effects.	Paediatric intensive care	Not stated	Two intensive care paediatricians, two clinical pharmacists and a nurse manager
Manrique-Rodriguez S. et al., 2012 [10]	Develop a drug library to help prevent serious medication errors occurring during intravenous administration	Development of a drug library for IV drugs that were commonly used or classified as high-risk or that posed issues with administration	Compliance was 85%. In total, 94% of PICU nurses would recommend implementing this technology in other units. Several potential harmful infusion-related programming errors were prevented	11-bed paediatric intensive care unit	Yes	Clinical pharmacist, PICU paediatrician and chief nurse for the unit
Namshirin P. et al., 2011 [27]	Commentary/descriptive article	Selection of a suitable device—formalising human factors analysis in the purchasing protocol for medical devices	Team was successful in choosing an infusion pump with DERS that met the needs of all stakeholders	Ambulatory	Not stated	Anaesthesiology, nursing, pharmacy purchasing, biomedical engineering quality and patient safety and human factors experts
Walroth T.A. et al., 2018 [17]	Reduce clinically insignificant alerts from smart infusion pumps	Development and implementation of an inter-professional consensus to review and optimise the drug library and dosing limits	Review of total number of alerts per smart pump revealed a 50% decrease in the median number of alerts per device over a 4-year period.	6 health systems	Yes	Pharmacists and nurses from each of the 6 representative health systems and industrial engineers
Wetterneck T.B. et al., 2006 [18]	Identify risks of implementation of smart pumps and evaluate IV pump technologies that could improve pump programming accuracy and decrease errors with IV medication	Failure mode and effects analysis (FMEA) was used to guide successful implementation	FMEA is a useful tool for implementation of smart pumps with DERS. Further refinements were required for paediatric concentrations	Hospital—a tertiary care, academic medical centre	Yes, for adults only	Anaesthesiology physicians, equipment, biomedical and industrial engineers, internal medicine, nursing, pharmacy, and quality improvement carried out FMEA. Drug library developed by pharmacists, nurses, an anaesthesia engineer and physicians.

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
