# Peer review of "Implementation of Smart Infusion Pumps: A Scoping Review and Case Study Discussion of the Evidence of the Role of the Pharmacist"

_pharmacy, 2020, doi:10.3390/pharmacy8040239_

Round 1
Reviewer 1 Report
Nice introduction describing the evolvement of pumps, injection medication errors, drug error reduction software, the role of the pharmacist, and the aims of the paper. Overall nice literature review and summary including the case study. The case study is also detailed enough that others could duplicate.
I believe there might be another paper in the future associated with this project - consider a follow-up manuscript on lessons learned at this site once implemented and used for a substantial amount of time.
I suggest that there be a Conclusion section to this manuscript. The manuscript has a discussion that includes remarks appropriate for a conclusion – I suggest this be considered.
There are a few words missing throughout – here are some examples – the entire article should be re-read and checked for such things:
- Line 217 – change “in” to “a”
- Line 223 – i.e., nursing staff - should be in parenthesis
- Line 224 – medication should be medications (plural)
- Line 227 – e.g., instead of the rate being present within the prescription as ml/hr - should be in parenthesis
Writing style is very good. There are several British spelling “errors” that should be corrected - this includes but is not limited to words such as: organisations, paediatric, realising, emphasise, recognised, focussing, standardise, minimise, etc,. The authors should carefully check the entire paper including Table 1 for needed spelling corrections.
Table 1 should be referenced in the body of the article. It is just attached at the end.
Table 1. This table needs to be completely re-formatted. On my copy, it took up 17 pages when printed. It is very difficult to read and utilize in the current format. The double spacing, use of columns, and both right and left justification of columns make this important table un-useable. Re-think how this information can be presented in a much more user friendly table. Consider putting information in rows for each reference? Also, is there an organized method for how the articles are listed – for example, by themes? I couldn’t tell. Thank you for your work on improving this portion of the article.
Author Response
Reviewer 1:
Nice introduction describing the evolvement of pumps, injection medication errors, drug error reduction software, the role of the pharmacist, and the aims of the paper. Overall nice literature review and summary including the case study. The case study is also detailed enough that others could duplicate.
Thank you for the positive feedback
I believe there might be another paper in the future associated with this project - consider a follow-up manuscript on lessons learned at this site once implemented and used for a substantial amount of time.
Thank you for the recommendation of a follow up manuscript, we hope to action this in due course.
I suggest that there be a Conclusion section to this manuscript. The manuscript has a discussion that includes remarks appropriate for a conclusion – I suggest this be considered.
We have added a conclusion section as follows.
With the implementation of any new health technology, the organisations should involve the key stakeholders. In the case of smart infusion pumps, pharmacists have a role at all stages from the tendering and procurement process through to implementation and evaluation. Standardisation of intravenous infusions should be considered a pre-requisite to building a drug library for smart infusion pumps, and sufficient time and resources should be allocated for the creation of the drug library in order for it to be accurate and safe.
There are a few words missing throughout – here are some examples – the entire article should be re-read and checked for such things:
- Line 217 – change “in” to “a”
- Line 223 – i.e., nursing staff - should be in parenthesis
- Line 224 – medication should be medications (plural)
- Line 227 – e.g., instead of the rate being present within the prescription as ml/hr - should be in parenthesis
Thank you for noting the grammatical and typographical errors. We have addressed the ones highlighted above, re-checked the entire article, and make corrections as necessary. These are not detailed here, but evident in the document through tracked changes functionality.
Writing style is very good. There are several British spelling “errors” that should be corrected - this includes but is not limited to words such as: organisations, paediatric, realising, emphasise, recognised, focussing, standardise, minimise, etc,. The authors should carefully check the entire paper including Table 1 for needed spelling corrections.
Thank you for noting the inconsistencies. We have checked the entire paper, corrected misspellings, and used UK English consistently through the article.
Table 1 should be referenced in the body of the article. It is just attached at the end.
Table 1 is referenced in the results (section 3, sentence 2) and is now placed in the main text near to the first time it is cited.
Table 1. This table needs to be completely re-formatted. On my copy, it took up 17 pages when printed. It is very difficult to read and utilize in the current format. The double spacing, use of columns, and both right and left justification of columns make this important table un-useable. Re-think how this information can be presented in a much more user friendly table. Consider putting information in rows for each reference? Also, is there an organized method for how the articles are listed – for example, by themes? I couldn’t tell. Thank you for your work on improving this portion of the article.
The formatting of table 1 has been revised to make it more user-friendly and arranged in alphabetical order by author name.
Reviewer 2 Report
Dear Authors
After reading the manuscript, I understand that this is a scoping review and not a systematic review. This fact is important and must be mentioned in the title. In addition, the limitations of a scoping review should be mentioned in a paragraph. Authors should read "Systematic review or scoping review? Guidance for authors when choosing between a systematic or scoping review approach" By Munn, Z et al. In the case of a systematic review the procedure described by the authors cannot be taken for granted.
Kind Regards
Reference
Munn, Z., Peters, M.D.J., Stern, C. et al. Systematic review or scoping review? Guidance for authors when choosing between a systematic or scoping review approach. BMC Med Res Methodol 18, 143 (2018). https://doi.org/10.1186/s12874-018-0611-x
Author Response
After reading the manuscript, I understand that this is a scoping review and not a systematic review. This fact is important and must be mentioned in the title. In addition, the limitations of a scoping review should be mentioned in a paragraph. Authors should read "Systematic review or scoping review? Guidance for authors when choosing between a systematic or scoping review approach" By Munn, Z et al. In the case of a systematic review the procedure described by the authors cannot be taken for granted.
This is indeed a scoping review and not a systematic review. We have, as advised, revised the title, included a statement in the methods section and mentioned the limitations of a scoping review in the discussion section as shown below.
Title: "Implementation of smart infusion pumps: a scoping review of the evidence of the role of the pharmacist."
Method: A scoping literature review was conducted using Embase and Ovid Medline
Discussion: the penultimate paragraph contains the following statements:
We used a scoping review to gain an overview of the evidence of the role of the pharmacist in implementing smart infusion pumps. ……….. A limitation of the scoping review approach is that it cannot not provide answers to specific questions. For example, in this review the evidence of the exact role and impact of the pharmacists’ involvement in specific areas such as training cannot be synthesised.
Reviewer 3 Report
The use of smart infusion pumps, improve safety, but there are many consideration to take into account, in order that this could be guaranteed.
Therefore, it is an interesting manuscript, even though; there are some issues to consider:
- The aim of the manuscript is to focus on the implication of pharmacists in the process of the implementation. However, over the text, not always is this the focus. This is not surprising as it is a review of publications with very different scope, and it is a process where different healthcare professionals are involved. Therefore, I would suggest that the title of the manuscript do no stress so this issue. I suggest for example:
Implementation of smart infusion pump. Evidence of the role of the pharmacist.
- A very important point is the standardization. I think that it has to be taken into account if there was a standardization of the IV admixtures, before implementing the smart infusion pumps. Is it this point explained in the publications reviewed? I consider that the process of standardization would be easier, safer and less time consuming if previously the hospital has protocols of IV administration.
- Line 181.For me this sentence is not clear. Were not set previously, the infusion pumps in millilitres/hour? It is the usual set for rate infusion.
- All along the text, there is much explanation about dosing, rates. However, what about drug compatibility? How is approached this problem when different drugs are infused at the same time?
- Discussion, paragraph beginning in line 269. The authors discuss that literature shows that pharmacist were key stakeholder in the implementation of drug libraries, due that they are drug experts. This is certainly true, but also I would add that pharmacists in they everyday work are constantly doing interventions and giving advice regarding drug admixtures and IV administration, therefore is an inherent part of their job, and have the tools and competencies to do it.
Author Response
The use of smart infusion pumps, improve safety, but there are many consideration to take into account, in order that this could be guaranteed.
Therefore, it is an interesting manuscript, even though; there are some issues to consider:
- The aim of the manuscript is to focus on the implication of pharmacists in the process of the implementation. However, over the text, not always is this the focus. This is not surprising as it is a review of publications with very different scope, and it is a process where different healthcare professionals are involved. Therefore, I would suggest that the title of the manuscript do no stress so this issue. I suggest for example:
Implementation of smart infusion pump. Evidence of the role of the pharmacist.
Thank you for the positive feedback. We agree this is an interesting area involving a range of health professionals. We have revised the title as advised to: "Implementation of smart infusion pumps: a scoping review of the evidence of the role of the pharmacist."
- A very important point is the standardization. I think that it has to be taken into account if there was a standardization of the IV admixtures, before implementing the smart infusion pumps. Is it this point explained in the publications reviewed? I consider that the process of standardization would be easier, safer and less time consuming if previously the hospital has protocols of IV administration.
We agree standardisation is very important. This is now emphasised in the results (first sentence of section 3.2.3): ‘The two predominant themes for the creation of a drug library were using a multidisciplinary team, consisting of a pharmacist, nurses and physicians and standardisation of infusions.’
and with a new column in table 1 to indicate which of the publications reviewed involved the use of standardised infusions.
Line 181.For me this sentence is not clear. Were not set previously, the infusion pumps in millilitres/hour? It is the usual set for rate infusion.
We have revised the paragraph to clarify this point: ‘The pre-existing infusion pumps required the rate to be set manually in millilitres/hour (ml/hr). The drug library set up allowed the display to default to rate (ml/hr) or dose (mg/kg/min). The clinical advisory group was consulted about this as it required a change in work practices and introduced a potential risk in misinterpreting the display value. The group advised that for safety reasons, the dosing units and rate display should match the prescription as much as possible.’
- All along the text, there is much explanation about dosing, rates. However, what about drug compatibility? How is approached this problem when different drugs are infused at the same time?
Drug compatibility issues are an important aspect of infusion therapy but are not likely to be implicated by dose error reduction software, therefore this was not discussed within the scope of the review.
- Discussion, paragraph beginning in line 269. The authors discuss that literature shows that pharmacist were key stakeholder in the implementation of drug libraries, due that they are drug experts. This is certainly true, but also I would add that pharmacists in they everyday work are constantly doing interventions and giving advice regarding drug admixtures and IV administration, therefore is an inherent part of their job, and have the tools and competencies to do it.
We agree and have strengthened this point in paragraph two of the discussion section: ‘Pharmacists, as medicines experts, are best placed to create the drug library as they have the working knowledge of how these medicines are used, the evidence behind their use, and the concentrations required. Advising on how medications should be prepared and administered is inherent to a pharmacist’s role and therefore they have the necessary expertise and competencies.’
Reviewer 4 Report
Thank you for the submission. It is well-written, albeit a very bland topic. I am not sure how much interest there will be amongst the journal’s readers. Given the topic (including the involvement of pharmacists as a key outcome), one would expect to see the International Pharmaceutical Abstracts (IPA) searched amongst the databases.
How were the key themes determined? It all seems very arbitrary and requires a more scientific basis. The text in the table should be reduced. The table should be much more concise.
Author Response
Thank you for the submission. It is well-written, albeit a very bland topic. I am not sure how much interest there will be amongst the journal’s readers. Given the topic (including the involvement of pharmacists as a key outcome), one would expect to see the International Pharmaceutical Abstracts (IPA) searched amongst the databases.
How were the key themes determined? It all seems very arbitrary and requires a more scientific basis. The text in the table should be reduced. The table should be much more concise.
The method section has been revised to explain that ‘Themes that featured prominently in the included literature were identified using a deductive thematic approach’.
The text in the table has been reduced and made more concise.
Round 2
Reviewer 2 Report
D
Dear Authors
There are differences between a Scoping Review and a Systematic Review. These differences are based on 5 points: Hypothesis, Objective, Type of studies included, Results and Inclusion criteria. These differences although they influence the search equations, and therefore it is necessary to describe the processes for obtaining the most relevant studies.
The present study lacks an explanation of the methodology used and it is practically impossible for the reviewer to approximate the search carried out by the authors. I believe that the authors should place themselves in the reader's position and make a considerable effort to explain the methodology, explaining from the search equation used, a good definition of the inclusion criteria as well as a correct definition of the process.
The meaning of the last paragraph of the methodology section is not understood.
Author Response
Reviewer 2 comments:
There are differences between a Scoping Review and a Systematic Review. These differences are based on 5 points: Hypothesis, Objective, Type of studies included, Results and Inclusion criteria. These differences although they influence the search equations, and therefore it is necessary to describe the processes for obtaining the most relevant studies.
We have included a hypothesis leading to the objective as follows: ‘Our hypothesis was that whilst studies highlight that the drug library build should be undertaken by a multidisciplinary team, including a pharmacist, few have reported the extent or nature of the input required by the pharmacist for greatest benefit. Therefore, the primary objective of this paper was to review the literature to identify key factors for the implementation of the smart infusion pumps, with a focus on the role of pharmacists’
The present study lacks an explanation of the methodology used and it is practically impossible for the reviewer to approximate the search carried out by the authors. I believe that the authors should place themselves in the reader's position and make a considerable effort to explain the methodology, explaining from the search equation used, a good definition of the inclusion criteria as well as a correct definition of the process.
We have revised the method section as advised and included the search string/equation and elaborated on the inclusion criteria.
A scoping literature review was conducted using Embase and Ovid Medline (dates from inception to 13th July 2020). A review protocol was developed by the authors but not registered. The same search strategy used for all the databases, using key words as follows: (“smart pump” OR “smart pumps” OR “intravenous smart pumps” OR “smart IV pump” OR “smart pump safety” OR “smart pump technology”) AND (“drug libraries” OR “drug library” OR “drug error reduction software”) AND (“pharmacist” OR “pharmacy”) AND (“implement*” and “develop*”). Articles were restricted to the English language. All duplicates were removed. Inclusion criteria were any original research studies, narrative papers or abstracts detailing or describing the process of implementation of the smart pumps or the process of creating the drug library. … … descriptive information including the study objective, reported intervention, reported study outcomes, study setting, whether standardised concentration infusions were used and details of the healthcare professionals involved in the implementation and development of the drug library was collated into a table.
The meaning of the last paragraph of the methodology section is not understood.
The final paragraph of the methods is an explanation of the case study comparison using the themes identified from the review.
Yours sincerely,
Neha Shah and Yogini Jani
